# Evaluation of Polymer-Coated Carbon Nanotube Flexible Microelectrodes for Biomedical Applications

**DOI:** 10.3390/bioengineering10060647

**Published:** 2023-05-26

**Authors:** Chethani Ruhunage, Vaishnavi Dhawan, Chaminda P. Nawarathne, Abdul Hoque, Xinyan Tracy Cui, Noe T. Alvarez

**Affiliations:** 1Department of Chemistry, University of Cincinnati, Cincinnati, OH 45221, USA; ruhunack@mail.uc.edu (C.R.); nawarawp@mail.uc.edu (C.P.N.); hoqueml@mail.uc.edu (A.H.); 2Department of Bioengineering, University of Pittsburgh, Pittsburgh, PA 15261, USA; vad33@pitt.edu

**Keywords:** carbon nanotubes, flexible polymer coatings, pinhole free polymer coatings, hydrogenated nitrile butadiene rubber

## Abstract

The demand for electrically insulated microwires and microfibers in biomedical applications is rapidly increasing. Polymer protective coatings with high electrical resistivity, good chemical resistance, and a long shelf-life are critical to ensure continuous device operation during chronic applications. As soft and flexible electrodes can minimize mechanical mismatch between tissues and electronics, designs based on flexible conductive microfibers, such as carbon nanotube (CNT) fibers, and soft polymer insulation have been proposed. In this study, a continuous dip-coating approach was adopted to insulate meters-long CNT fibers with hydrogenated nitrile butadiene rubber (HNBR), a soft and rubbery insulating polymer. Using this method, 4.8 m long CNT fibers with diameters of 25–66 µm were continuously coated with HNBR without defects or interruptions. The coated CNT fibers were found to be uniform, pinhole free, and biocompatible. Furthermore, the HNBR coating had better high-temperature tolerance than conventional insulating materials. Microelectrodes prepared using the HNBR-coated CNT fibers exhibited stable electrochemical properties, with a specific impedance of 27.0 ± 9.4 MΩ µm^2^ at 1.0 kHz and a cathodal charge storage capacity of 487.6 ± 49.8 mC cm^−2^. Thus, the developed electrodes express characteristics that made them suitable for use in implantable medical devices for chronic in vivo applications.

## 1. Introduction

Achieving stable electrode–tissue interfaces remains among the most challenging tasks in the development of long-lasting implantable electrodes for chronic applications. Implantable electrode devices such as neural electrodes can provide detailed information about brain activity and disorders by recording real-time brain signals [1]. Implantable neural electrodes can also be used to stimulate specific areas of neural tissues as a therapeutic strategy to treat neurological disorders such as Parkinson’s disease, Alzheimer’s disease, and epilepsy [2]. Owing to the chronic nature of many neurological disorders, it is critical that implantable electrodes exhibit long-term performance stability to allow continuous treatment and ensure signal accuracy [3,4]. Unfortunately, the life span of neural electrodes is limited by various biotic and abiotic factors [4,5]. In particular, inflammatory host tissue responses are a main contributor to the performance deterioration of neural electrodes over time [6,7,8]. Stiff materials such as silicon and metals cause significant inflammatory responses owing to their mechanical mismatch with soft tissues [7,9,10]. Among the approaches available for developing stable neural interfaces, fabricating subcellular sized probes using flexible and soft materials is promising [11,12].

Although electrode miniaturization can minimize inflammatory responses, electrochemical properties such as impedance, charge storage capacity (CSC), and charge injection limit (CIL) are negatively affected. As conductive nanomaterials, carbon nanotubes (CNTs) have been used as coatings on metal electrodes to improve electrochemical performance [13,14,15]. In addition, CNT-based neural electrodes have provided improved mechanical compatibility with soft tissues and reduced inflammatory responses [9,10]. More recently, CNTs have been used independently to develop neural electrodes. CNTs can be easily assembled into self-standing CNT fibers with diameters of less than 10 µm [10,16]. CNT fibers are softer and more flexible than rigid metal- and silicon-based neural electrodes and have a bending stiffness that is orders of magnitude lower than those of silicon and carbon fiber electrodes [17,18]. Owing to their unique properties, CNTs can facilitate the generation of stable neural interfaces [8,13,19,20,21].

In addition to the conductive components of neural electrodes, insulating materials play a significant role in establishing stable neural interfaces. Packaging materials isolate internal electronics from surrounding fluids by preventing the permeation of water vapor and ions [2,22,23]. To ensure performance stability, neural electrodes require coatings with chemical and mechanical stability under physiological conditions. It has been hypothesized that flexible polymer coatings on flexible CNT fibers would minimize the mechanical mismatch between electrodes and soft tissues, thus suppressing inflammatory responses and generating neural interfaces that are more compatible with brain tissues [22,24,25].

Among electrode-insulating polymers commonly employed in neural prostheses, parylene C, polyimide, and polydimethylsiloxane (PDMS) are relatively soft coating materials [22,26,27,28]. In addition, liquid crystal polymers have received recent interest as soft coating materials for neural electrodes [29,30]. However, thin PDMS coatings are not favorable for insulating ultrasmall neural probes due to high permeability and delamination issues [23,31,32]. Although parylene C itself exhibits excellent properties, such as chemical inertness, high purity, low moisture absorption, conformal deposition, and biocompatibility [33,34], most of the properties of parylene C films depend on the polymer deposition method, i.e., chemical vapor deposition (CVD) [33]. Parylene C coatings suffer poor integration with many substrates, which allows water vapor to accumulate over time [35,36,37]. In addition, the brittleness of parylene C and its tendency to crack upon bending are disadvantageous for insulating neural electrodes. Polyimide exhibits high-temperature stability, low moisture absorption, and mechanical durability, but the limited insulation lifetime of polyimide coatings in saline environments (2–7 years) may restrict chronic implantation [23,28,31,38,39]. Therefore, there is a critical need for insulating polymer coatings with chemical and mechanical stability as well as flexibility that can be seamlessly integrated with substrates. Moreover, in terms of polymer fabrication techniques, it is critical that conformal, uniform, and pinhole-free dense barriers can be produced [23,40].

Hydrogenated nitrile butadiene rubber (HNBR) is a flexible polymer with excellent mechanical properties and abrasion, temperature, and chemical resistance under extreme conditions [41,42,43]. Due to its polyfunctionality, HNBR has been extensively used in oilfield applications, where long-term stability is essential under extreme conditions [44]. Therefore, it has been hypothesized that HNBR is an excellent candidate for insulating medical implants that require long-term stability. Recently, our lab showed that HNBR can be used to insulate CNT fibers via a continuous dip-coating (CDC) approach to generate electric microcables [43]. In the current study, the CDC technique was employed to fabricate HNBR-coated CNT fibers for the development of flexible neural electrodes. Various CDC parameters, including the solvent, polymer concentration, and withdrawal speed, were evaluated to achieve pinhole-free and uniform HNBR coatings. The applicability of the CDC technique to CNT fibers with a wide range of diameters (25–66 µm) and a length up to 4.8 m was demonstrated. The morphology, chemical nature, and insulating properties of the HNBR-coated CNT fibers were analyzed. The biocompatibility of HNBR was evaluated in vitro using fibroblast and neuronal cultures as a prerequisite for in vivo studies. Furthermore, the electrochemical properties of microelectrodes fabricated from the HNBR-coated CNT fibers were evaluated. In addition, aging studies were conducted at elevated temperatures to evaluate the polymer stability for chronic applications. 

## 2. Experimental Section

### 2.1. Reagents and Materials

Vertically aligned CNT (VA-CNT) arrays were provided by Professor Vesselin N. Shanov (Nanoworld Laboratories, University of Cincinnati). HNBR (Zetpol 2000) was purchased from Zeon Chemicals (Louisville, KY, USA). Acetone (C_3_H_6_O, 99.7%), copper(II) sulfate (CuSO_4_·5H_2_O), 2,5-dimethyl-2,5-di(*tert*-butylperoxy)hexane (C_16_H_34_O_4_, 92%), Dulbecco’s modified Eagle medium (DMEM, Gibco, Grand Island, NY, USA), Neurobasal Plus medium, fetal bovine serum (FBS, Gibco), penicillin–streptomycin (pen/strep, Life Technologies, Carlsbad, CA, USA), and sodium 2,3-bis(2-methoxy-4-nitro-5-sulfophenyl)-5-[(phenylamino)-carbonyl]-2*H*-tetrazolium (XTT) cell viability assays (Invitrogen) were purchased from Thermo Fisher Scientific (Waltham, MA, USA). Anhydrous methylene chloride (CH_2_Cl_2_, 99.8%) and phosphate-buffered saline (PBS) were purchased from Sigma Aldrich (St. Louis, MO, USA). Silver paint was purchased from Electron Microscopy Sciences (Hatfield, PA, USA). NIH/3T3 fibroblast cells were purchased from ATCC (Manassas, VA, USA). Milli-Q water (18.2 MΩ cm) was used to prepare all reagent solutions.

### 2.2. CNT Fiber Synthesis and Densification

Spinnable VA-CNT arrays were utilized to fabricate CNT fibers. The synthesis of VA-CNT arrays by CVD and the subsequent fabrication of CNT fibers by dry-spinning have been reported by our group elsewhere [45,46]. Briefly, the dry-spinning process was initiated by drawing CNTs from the edge of the VA-CNT array into a film while simultaneously twisting to assemble a CNT fiber, as shown schematically in Figure 1A. A homemade twisting apparatus was used to spin the fiber continuously. As the freshly spun fiber was not densely packed, it was passed through acetone to increase the CNT density within the fiber. This densification process was also performed continuously using a homemade setup, where the CNT fiber on a bobbin was dipped into acetone and the CNT fiber was then withdrawn at a speed of 1.6 mm s^−1^ (Figure 1B). Densified CNT fibers with diameters of 25–66 µm were used in this study.

### 2.3. Fabrication of HNBR-Coated CNT Fibers Using the CDC Approach

In this study, the polymer used to coat the CNT fibers was Zetpol 2000 HNBR (Zeon Chemicals, Louisville, KY, USA). According to the manufacturer’s description, Zetpol 2000 HNBR is a fully saturated copolymer of butadiene and acrylonitrile (ACN) with 36% bound ACN content. Dichloromethane was used to disperse HNBR. The critical CDC parameters, such as HNBR concentration and withdrawal speed, were optimized to achieve complete insulation of the CNT fiber. The optimized conditions are described here. HNBR (1.7 g) was dispersed in 17 mL of dichloromethane by continuous stirring overnight to give a final concentration of 0.1 g mL^−1^. As a cross-linking agent, 2,5-dimethyl-2,5-di(*tert*-butylperoxy)hexane (170 mg) was added dropwise to the polymer solution and stirred for 5 min. Then, the polymer mixture was degassed under vacuum to remove air bubbles. A homemade dip-coating apparatus was used to continuously coat CNT fibers with HNBR (Figure 1C). The densified CNT fiber was initially collected on a spool that could also be used as the delivering spool. From the delivering spool, the CNT fiber was drawn horizontally through a bath containing HNBR, passed through a miniature cylindrical furnace (160 °C) to ensure cross-linking of the HNBR coating, and then collected on a spool (9 mm diameter) at a constant speed of 5 mm s^−1^. 

### 2.4. Polymer Coating Evaluation

High-resolution scanning electron microscopy (SEM) images were obtained using an FEI Apreo (Thermo Scientific, Waltham, MA, USA) or TM4000 (Hitachi, Santa Clara, CA, USA) scanning electron microscope at an acceleration voltage of 5 kV to analyze the polymer-coated CNT fibers and CNT fiber cross sections. 

### 2.5. Spectroscopic Analyses

#### 2.5.1. X-ray Photoelectron Spectroscopy (XPS)

The chemical composition of HNBR was determined using XPS. For the XPS samples, glass slides (5 cm^2^) were used as the substrate instead of CNT fibers. The glass slides were manually dip-coated with HNBR (0.1 g mL^−1^) and then cured at 160 °C for 10 min. Glass slides without an HNBR coating were evaluated as a control. XPS measurements were performed using a K-Alpha X-ray photoelectron spectrometer (Thermo Scientific, Waltham, MA, USA) equipped with an Al Kα X-ray microfocused monochromator and a multichannel detector. High-resolution XPS spectra were collected for the C 1s, O 1s, N 1s, Cl 2p, and Si 2p core levels. An analyzer pass energy of 20 eV was used for all core level scans, and the photoelectron take-off angle was 90° with respect to the sample plane with a spot size of 400 µm. The XPS spectra were analyzed using Avantage surface chemical analysis software, and the core level spectra were deconvoluted using Origin Pro 8.5 software.

#### 2.5.2. Fourier Transform Infrared (FT-IR) and Raman Spectroscopy

HNBR-coated CNT fibers with an optimized coating thickness of 10 µm were analyzed using FT-IR and Raman spectroscopy. FT-IR spectra were recorded using a Nicolet 6700 FT-IR spectrometer (Thermo Scientific, Waltham, MA, USA) equipped with a germanium attenuated total reflectance (ATR) sampling module (Thermo Scientific, Waltham, MA, USA). Raman measurements were carried out using a Renishaw inVia Raman microscope (West Dundee, IL, USA) with an Ar-ion laser at an excitation wavelength of 633 nm. CNT fibers with no coating were analyzed as a control.

### 2.6. Fabrication of HNBR-Coated CNT Fiber Microelectrodes

Using a sharp razor blade, the HNBR-coated CNT fiber was cut into 2 cm pieces, which were used to make microelectrodes. Sectioning of the HNBR-coated CNT fiber exposed a cross section of the CNT fibers encapsulated within the polymer. The exposed cross section consisted of millions of individual CNTs within the CNT fiber, while the sidewalls of the CNT fiber assembly were coated with 10 µm thick HNBR. Silver paint was applied on one side of the polymer-coated CNT fiber cross section to electrically bridge the exposed CNTs and make an electrical connection to a copper wire. The copper/silver electrical connection was then sealed with an epoxy resin. The 2 cm long polymer-coated fiber was trimmed to 1 cm using a sharp razor blade to give a microelectrode consisting of the exposed CNT fiber cross section. A detailed schematic of the microelectrode fabrication process is shown in Appendix A. The fabricated microelectrode surface consists of millions of individual carbon nanotubes with exposed open ends (CNT tips).

### 2.7. Electrochemical Characterization

All electrochemical measurements were performed using a PalmSens4 electrochemical workstation (Houten, The Netherlands) with a three-electrode setup. The HNBR-coated CNT fiber microelectrode was used as the working electrode. A solid-state Ag/AgCl electrode and a Pt wire were used as the reference and counter electrodes, respectively. For all electrochemical experiments, five replicates were performed, unless otherwise mentioned.

#### 2.7.1. Electrical Insulating Properties of HNBR Coating

Complete insulation of the CNT fibers was achieved by optimizing the CDC parameters, specifically the HNBR polymer concentration and the withdrawal speed. The degree of electrical insulation for HNBR-coated CNT fibers prepared under various conditions was evaluated using SEM, electrochemical impedance spectroscopy (EIS), and electrodeposition. Electrodeposition testing was performed using electrodes fabricated as described in Section 2.6. Using the prepared electrode as the working electrode, chronoamperometric electrodeposition was performed in a 50 mM CuSO_4_ aqueous solution at a potential of −0.2 V for 10 s. The sidewalls of the HNBR-coated CNT fiber electrodes were then analyzed using SEM, as insufficient polymer coverage around the CNT fiber is expected to result in Cu particle deposition.

#### 2.7.2. EIS Analysis

EIS was used to evaluate the impedance of the HNBR-coated CNT fiber microelectrodes. EIS measurements were performed in 0.01 M PBS (pH 7.4) in the frequency range of 1 MHz to 0.1 Hz with an amplitude of 10 mV.

#### 2.7.3. Cyclic Voltammetry (CV)

CV in 0.01 M PBS (pH 7.4) was used to electrochemically pretreat and stabilize the HNBR-coated CNT fiber microelectrodes. The pretreatment was performed by cycling the potential from +1 V to −1 V at a scan rate of 0.1 V s^−1^ for 50 cycles (until a stable current was achieved). CV was also used to analyze the CSC and the water window of the fabricated microelectrodes. The CSC was analyzed in the potential range of +1 to −1 V at a scan rate of 0.1 V s^−1^. The water window was analyzed by cycling the potential from +2 to −2 V at a scan rate of 0.1 V s^−1^.

### 2.8. Long-Term Stability

The long-term stability of the HNBR-coated CNT fiber microelectrodes and the integrity of the HNBR coating were analyzed via electrochemical testing and SEM. The HNBR-coated CNT fiber microelectrodes were soaked in 0.01 M PBS at 37 °C for 4 weeks. Accelerated aging tests were also performed at 60 and 75 °C for 2 weeks. The effects of aging were evaluated using EIS and CV, and morphological changes were analyzed using SEM.

### 2.9. In Vitro Cytotoxicity of HNBR Coating

#### 2.9.1. Elution Test

To evaluate the toxicity of the HNBR coating, glass slides were coated with a 0.05 g mL^−1^ HNBR solution and cured at 160 °C for 10 min. The coated glass slides were cleaned by soaking them in autoclaved distilled water for 24 h followed by soaking in absolute ethanol for 30 min. The samples were submerged in 3 mL of serum-free DMEM/nutrient mixture F-12 (F12, Gibco) at 37 °C and 5% CO_2_. Glass slides without an HNBR coating were used as a control. A positive control was prepared using 0.1% Triton-X in 3 mL of DMEM/F12, and incubated DMEM/F12 acted as a negative control. Following 1 day of incubation, the elution media of the HNBR-coated samples and controls were supplemented with 10% FBS and 1% pen/strep. Separately, 3T3 fibroblast cells were seeded in a 48-well plate at a seeding density of −20,000 cells well^−1^ and grown in DMEM supplemented with 10% FBS and 1% pen/strep. The cells were incubated at 37 °C in a humidified environment of 5% CO_2_ until 80–90% confluency. The growth medium in each well was replaced with 100% serum-supplemented elution medium. The cells were incubated for 24 h and the cell viability was then assessed using an XTT assay. XTT was prepared at a concentration of 1 mg mL^−1^ in pre-warmed supplemented media. *N*-Methyl dibenzopyrazine methyl sulfate (PMS) prepared at a concentration of 10 × 10^−3^ M in sterile PBS served as the electron donor. Immediately prior to introduction to the cells, 10 µL of PMS solution was added to 4 mL of XTT solution. Then, XTT/PMS (62.5 µL) was added to each well, which contained 250 µL of medium. After incubation for 3 h, the resulting color changes were evaluated by measuring the absorbance at 450 nm using a spectrophotometer (SpectraMax i3, Molecular Devices). Three independent cell cultures were prepared, and the blank-subtracted absorbance values were normalized to the media-only negative control for each culture. The normalized cell viability represents the average of the three cultures and is plotted with error bars representing the standard deviation.

#### 2.9.2. Neuronal Cultures

Primary neurons were isolated from embryonic day 18 rat fetuses following previously published protocols [47]. Briefly, the mother rat was euthanized under CO_2_ followed by cervical dislocation and the pups were removed. From individual pups, cortices were isolated and submerged in 0.15% trypsin solution to obtain cells, which were then resuspended in neurobasal medium supplemented with B27 and pen/strep. Neurons were plated at 30,000 cells cm^−2^ in a 24-well plate containing HNBR-coated fibers and grown for 3 days before fixing with 4% paraformaldehyde. The fixed cells were stained to visualize neurite outgrowth (anti-β-III tubulin) and nuclei (DAPI). A confocal microscope (Olympus Fluoview 1000) was used to collect z-stack fluorescent images. ImageJ was used to flatten the z-stack and obtain a snapshot of the neurons growing near the HNBR-coated CNT fibers.

## 3. Results and Discussion

### 3.1. Optimization of the CDC Technique for Fabricating Flexible HNBR-Coated CNT Fibers

Dip-coating techniques are widely used to coat various substrates with polymer materials. In the present study, a continuous approach was adopted to enable the coating of meters-long CNT fibers with HNBR in a single step. The CDC technique facilitates the fabrication of uniform and pinhole-free insulated CNT fibers, which have utility in both micro- and macroscale applications. Advantageously, the CDC approach is efficient, generates less organic solvent waste, and provides control over the polymer coating thickness based on the polymer concentration/viscosity and withdrawal speed. Recently, this technique has been used to coat polymer optical fibers and hollow fiber membranes with cladding materials and polyvinyl alcohol, respectively [48,49]. Alvarez et al. first reported the fabrication of HNBR-coated CNT fiber microcables using the CDC approach with acetone as a solvent [43].

Herein, the CDC approach was optimized to generate pinhole-free, uniform HNBR coatings with a controlled thickness on CNT fibers. The CNT fibers were prepared by dry-spinning a VA-CNT array and subsequent fiber densification. The solvent-induced densification process generally improves the conductivity and mechanical properties of CNT fiber [43,50,51]. Initially, the CDC process was optimized using a densified CNT fiber with a diameter of 44 µm (Figure 2A). Subsequently, the optimized parameters were used to coat CNT fibers with diameters in the range of 25–66 µm to demonstrate the wide applicability of the CDC approach.

The theory of liquid film development on a substrate via dip-coating is well-established and has been further extended for fiber geometries. Generally, the thickness of the meniscus depends on multiple parameters, including the fiber length, withdrawal speed, and fluid properties such as viscosity and density [48,52,53]. In this study, the HNBR concentration and fiber withdrawal speed were evaluated as critical parameters to achieve complete fiber coverage with a uniform thickness. For dip-coating, a suitable solvent is necessary to obtain uniform polymer films. Volatile organic solvents with low surface tensions, such as alcohols, are favored for dip-coating applications [54]. We previously dispersed HNBR using acetone because of its low boiling point (56 °C) [43], but dispersion was limited at HNBR concentrations lower than 0.055 g mL^−1^. In the current study, methylene chloride was chosen as the dispersion solvent, as HNBR could be dispersed at a wider range of concentrations. In addition, the low surface tension and fast evaporation (boiling point: 39.6 °C) of methylene chloride were found to be beneficial for uniform polymer film development on CNT fibers. 

Using a CNT fiber with a diameter of 44 µm, the withdrawal speed (1.5–13.0 mm s^−1^) was first optimized at a constant HNBR concentration of 0.055 g mL^−1^. The CNT fiber sidewall coverage and insulation properties were evaluated using SEM imaging and electrochemical testing. Although continuous coating of the CNT fiber was achieved at a minimum withdrawal speed of 6 mm s^−1^, the coating thickness was not sufficient to provide complete insulation. As shown by the SEM image in Appendix A, the underlying CNT fiber features were visible when the coating thickness was less than 1 µm. Fundamental research on dip-coating techniques has suggested that the coating thicknesses can be increased by using withdrawal speeds below 0.1 mm s^−1^ or above 1 mm s^−1^ [54]. The coating was found to be discontinuous when the withdrawal speed was below 6 mm s^−1^, although withdrawal speeds below 1.5 mm s^−1^ were not evaluated in this study. At higher withdrawal speeds, a slight improvement in the coating thickness was observed. However, higher withdrawal speeds resulted in a shorter exposure time for polymer cross-linking at 160 °C. Therefore, to provide sufficient time in the cylindrical furnace for cross-linking, further optimization was performed at a withdrawal speed of 6 mm s^−1^.

Subsequently, the effects of the HNBR concentration (0.055–0.10 g mL^−1^) on coating uniformity and thickness were analyzed. A significant improvement in polymer thickness was achieved using 0.1 g mL^−1^ HNBR. As shown in Figure 2B–D, CNT fibers with various diameters (24.8 ± 0.7, 43.6 ± 1.3, and 66.0 ± 0.3 µm) were coated with HNBR using the optimized CDC conditions (0.1 g mL^−1^ HNBR dispersion in methylene chloride at a withdrawal speed of 6 mm^−1^). Corresponding cross-sectional images are presented in Figure 2E–G. Under the optimized conditions, the topographic features of the CNT fiber were completely covered, the diameter was uniform along the fiber length, and the fabricated coating was pinhole free. The average HNBR coating thickness was 7 µm for the CNT fibers with a diameter of 24.8 µm.

The presence of defects and the insulation behavior of the CNT fibers coated under the optimized conditions were analyzed using Cu electrodeposition (Appendix A). To evaluate the limitations of the CDC technique, a 4.8 m long CNT fiber was continuously coated under the optimized CDC conditions. Continuous coating of the entire fiber was achieved, and no defects were observed at either end of this fiber, as shown by the SEM images in Appendix A. These results demonstrate the efficiency of the CDC approach for fabricating large quantities of pinhole-free and uniform HNBR-coated CNT fibers using inexpensive materials and a homemade setup.

Strong polymer–electrode adhesion is critical for developing insulating coatings on implantable neural electrodes, and the development of suitable polymer materials and technologies is an ongoing research area. Polymers such as parylene C show good performance as insulating materials but lack strong adhesion with underlying substrates [55,56]. The importance of this phenomenon is discussed further in Section 3.4. SEM images at the interface between the HNBR coating and CNT fiber (Appendix A) reveal the excellent wetting quality of the polymer. The SEM image in Appendix A was taken at the point on the CNT fiber where the polymer film began to develop at the meniscus. Appendix A shows a high-magnification SEM image of the HNBR-coated CNT fiber cross section. These images indicate seamless integration between HNBR and the CNT fiber, in agreement with our previous study [43], which is expected to be beneficial for developing neural electrodes. To further evaluate the suitability of the HNBR-coated CNT fibers for neural electrode development, the chemical, electrochemical, aging, and biocompatibility characteristics were evaluated.

### 3.2. Spectroscopic Characterization of HNBR Coating

To analyze the chemical nature of HNBR, the CNT fibers before and after coating were characterized using FT-IR and Raman spectroscopy. However, XPS was performed on HNBR-coated glass slides, as the HNBR-coated CNT fibers are much smaller than the instrumental limitation (lowest surface area of 400 µm diameter spot size).

Raman spectroscopic analysis of the uncoated CNT fiber with a diameter of 24.8 µm revealed the presence of characteristic D (sp^3^ carbon, 1330 cm^−1^), G (sp^2^ carbon, 1590 cm^−1^), and G′ (2660 cm^−1^) bands (black line, Figure 3A). The Raman spectrum of the fiber after coating contained additional peaks at 1080, 1442, 2230, 2857, and 2900 cm^−1^ (red line, Figure 3A). Similar peaks were observed for pristine Zetpol 2000 HNBR (blue line, Figure 3A). The peaks at 1442 and 2857–2900 cm^−1^ were attributed to the deformation and stretching vibrations, respectively, of CH_2_ groups in the highly saturated nitrile polymer [57,58,59]. The presence of nitrile groups was clearly indicated by the sharp peak at ~2230 cm^−1^, which was assigned to the v(C≡N) stretching vibration [57]. Typically, nitrile butadiene rubbers also exhibit peaks at ~1300 and 1640–1660 cm^−1^ (C=C stretching). However, as Zetpol 2000 HNBR is a completely hydrogenated nitrile polymer, the peak observed near 1590 cm^−1^ for the HNBR-coated fiber is likely due to the characteristic G band of the underlying CNT fiber [57]. This assignment was further confirmed by the absence of a peak at 1590 cm^−1^ for pristine Zetpol 2000 HNBR. The FT-IR spectrum of the coated CNT fiber also exhibited characteristic peaks corresponding to HNBR (Figure 3B) [60,61,62]. The peaks at 2926, 2856, and 1465 cm^−1^ were attributed to the stretching and deformation vibrations of CH_2_ groups in the polymer structure. The peak at 724 cm^−1^ was assigned to –CH_2_– vibrations in saturated C–C bonds. The distinct peak at 2237 cm^−1^ was attributed to –C≡N groups. These observations indicate that the chemical nature of HNBR was preserved after the coating process.

XPS was used to further confirm the chemical nature of HNBR. As shown by the XPS survey scan of an HNBR-coated glass slide (Figure 3C), the C 1s and N 1s peaks were predominant with a low-intensity O 1s peak. The atomic percentages of C, N, and O were 90.50 ± 0.67%, 7.49 ± 0.21%, and 1.66 ± 0.46%, respectively. The absence of a Cl peak confirmed that the solvent evaporated completely during withdrawal of the substrate. High-resolution N 1s, C 1s, and O 1s spectra are shown in Figure 3D–F. In the N 1s spectrum (Figure 3D), a single peak was observed at 399.9 eV corresponding to free –C≡N groups [62,63]. Deconvolution of the C 1s spectrum (Figure 3E) gave two peaks at 284.3 and 286.7 eV. According to the manufacturer, Zetpol 2000 HNBR is a fully hydrogenated nitrile rubber with little to no double bonds in the backbone. Therefore, the peak at 284.5 eV was assigned to C–C and C–H (sp^3^ C) in the polymer backbone [64,65]. The peak at 286.7 eV was assigned to –C≡N groups and C–O surface functional groups introduced after peroxide curing [62,64,65]. Deconvolution of the O 1s spectrum (Figure 3F) resulted in a single peak at 532.8 eV, which confirmed the introduction of C–O-containing functional groups during peroxide curing. Similar C–O-containing groups have been previously reported following nitrile rubber curing processes [64].

### 3.3. Cytotoxicity Evaluation of HNBR Coating

Implantable devices should be able to function in vivo without triggering undesirable immune and inflammatory responses [66,67]. Accordingly, biocompatibility is a key requirement for the clinical use of implantable biomaterials. Different aspects of biomaterials, such as their chemical nature, mechanical properties, and structural properties, can interact with the biological environment of the host and negatively influence the host tissue [67], resulting in undesirable effects, including cytotoxicity, sensitization, irritation, systemic toxicity, subchronic toxicity, genotoxicity, and inflammatory tissue responses. The biocompatibility of CNT fibers has been previously demonstrated [68,69,70]. Thus, in the current study, we focused on evaluating the cytotoxicity of the HNBR coating prepared from Zetpol 2000 HNBR. The manufacturer (Zeon Chemicals) states that Zetpol 2000 HNBR meets all FDA guidelines for safe and direct contact with food materials, but its grade has not been evaluated for implantable biomedical devices. Therefore, it is important to investigate the biocompatibility of HNBR.

Elution tests were primarily used to evaluate the toxicity of leachable materials such as unreacted raw materials, cross-linking agents, and impurities, which can leach into the local environment over time and generate toxic effects. After soaking HNBR-coated glass slides in serum-free DMEM for 1 day, the elution media was collected, treated with serum and antibiotics, and added to 3T3 fibroblasts cells after they reached 80–90% confluency. As shown in Figure 4A, the elution media from HNBR-coated samples did not have a toxic effect on the normalized cell viability, as compared to the positive control (one-way ANOVA, Kruskal–Wallis multiple comparison test, *** *p* < 0.001). Further, there was no significant difference between the HNBR, media-only, and glass-only control conditions, confirming the absence of toxic leachable materials in the HNBR coating.

We also performed a direct contact assay by placing HNBR-coated CNT fibers on cultured primary neurons. As visualized in Figure 4B,C, neurons grew seamlessly around and on the fiber itself. This observation further confirms the suitability of HNBR insulation for CNT fiber electrodes for neural applications.

### 3.4. Electrochemical Analysis of HNBR-Coated CNT Fiber Microelectrodes

#### 3.4.1. EIS Analysis

EIS is widely used to evaluate the properties of implantable microelectrodes including electrode–electrolyte interactions, metal electrode corrosion, conductive polymer oxidation, and electrode biofouling by proteins and inflammatory cells. Insulating polymers can also be subjected to delamination, degradation, and mechanical damage during insertion [35,71], which can expose the underlying conductive materials, resulting in device failure and health risks.

In the current study, EIS was used to investigate the stability of the HNBR-coated CNT fiber microelectrodes in vitro as an initial indicator of their suitability for its use in vivo. For the electrochemical evaluation, HNBR-coated CNT fiber microelectrodes were prepared using the 24.8 µm CNT fibers with an HNBR coating thickness of 7 µm. Prior to EIS analysis, the HNBR-coated CNT fiber microelectrodes were electrochemically pretreated to achieve stable performance. In particular, consistent charging currents were obtained after 50 CV cycles in the range of +1 to −1 V in PBS (pH 7.4), as shown in Appendix A. Electrochemical pretreatment significantly reduced the impedance of the microelectrodes, especially in the mid- and low-frequency ranges (Figure 5A). The phase angle and the Nyquist plot of a single microelectrode is shown in Appendix A, respectively. Figure 5B shows a comparison of the impedance at 1 kHz for HNBR-coated CNT fiber microelectrodes before and after electrochemical pretreatment (n = 15). The deviation in impedance was small after electrochemical pretreatment, with an average impedance of 55.9 ± 19.5 kΩ for electrochemically pretreated HNBR-coated CNT fiber microelectrodes at 1 kHz.

The stabilized electrodes were evaluated via SEM and electrodeposition to confirm that no physical damage/delamination occurred during electrochemical pretreatment. No physical damage to the HNBR coating was observed following repeated CV cycling. Electrodeposition in CuSO_4_ solution was used to investigate potential breakage of the polymer resulting from the application of higher voltages. SEM images of the side of the microelectrode and the microelectrode active area after electrodeposition are shown in Appendix A. Cu particle deposits were only observed on the microelectrode active area, not the sidewalls of the HNBR-coated CNT fiber microelectrodes, which implies that no polymer delamination occurred. Therefore, the significant improvement in impedance after electrochemical pretreatment was attributed to activation of the electrode surface area by increasing electrolyte penetration into the CNT fiber to form a double layer (activation-enhanced length effect) and surface functionalization by oxygen-containing functional groups. The activation-enhanced length effect of VA-CNTs has previously been studied experimentally and using equivalent circuit modeling [72]. Similar electrochemical pretreatment techniques are frequently used to activate carbon-based microelectrodes and are considered to introduce oxygen-containing functional groups and remove impurities [73]. The microelectrode consist of millions of densified CNTs with their tips and sidewalls exposed at the surface, which can undergo activation due to tip and sidewall functionalization. Depending on the density of CNTs and tip/sidewall contribution, the electrochemical properties could vary between samples. The calculated specific impedance of the stabilized microelectrode at 1 kHz was 27.0 ± 9.4 MΩ µm^2^, which is remarkable compared to those of previously reported metal- and carbon-based microelectrodes [9]. The low impedance of the fabricated HNBR-coated CNT fiber microelectrodes is favorable for both stimulation and recording purposes. As the long-term stability of both the HNBR coating and electrode–electrolyte interface is important for chronic applications, additional aging studies were performed, as discussed in Section 3.4.3.

#### 3.4.2. CSC and Water Window Analysis

The CSC is another important electrochemical property for neural electrodes. Neural interfaces with larger CSC values are preferential to enhance the performance of stimulation electrodes [23,74,75]. Delamination and electrolyte leakage through insulating coatings can lead to undesirable large electrical currents that can harm tissues and cells. Thus, monitoring the CSC provides information about insulating polymer failure as well as the stability of the electrode material. CSC, which is defined as the total available reversible charge per unit of geometrical surface area of the electrode, is typically estimated by performing CV in PBS within the water window [74,76,77,78]. The cathodal CSC (CSC_c_), which is frequently used to estimate the total reversible charge available in the cathodic phase of a pulse of stimulation area, was used in the current study. Owing to their extremely high surface areas and excellent conductive properties, CNTs are widely used to increase the CSC of metal electrodes [79].

To evaluate CSC_c_, the CNT fiber microelectrodes with a fiber diameter of 24.8 µm and an HNBR coating thickness of 7 µm were used. The water window for the HNBR-coated CNT fiber microelectrodes was estimated using CV in PBS (Appendix A). A water window in the range of −1.6 to 1.3 V was observed, which is much wider than those reported for metallic (Pt, Au, IrO_x_) and conductive-polymer-based neural electrodes [80,81,82,83]. Vitale et al. also reported a wide water window of −1.5 to 1.5 V for CNT fiber electrodes [9]. The CSC_c_ was calculated from the time integral of the cathodic current between potentials of −1 and 1 V (Figure 5C). The average CSC_c_ (487.6 ± 49.8 mC cm^−2^) was significantly larger than those of reported metal- and CNT-fiber-based neural electrodes [9,10]. The stabilized microelectrodes were monitored overnight to confirm the stability of CV and the EIS data (Appendix A).

#### 3.4.3. Aging Studies of HNBR-Coated CNT Fiber Electrodes

In vitro aging studies provide information about the durability of a material and its suitability for chronic applications. Prolonged microelectrode soaking may result in water vapor diffusion through the polymer, delamination due to weak adhesion, and unwanted reactions between the coating material and surrounding PBS solution, which can cause insulating coating materials to fail over time. The resulting exposure of underlying CNT fiber sidewalls can significantly increase charge transfer, which could be harmful. Accelerated aging tests at elevated temperature allow polymer degradation to be evaluated in a short period of time [23,84]. Accelerated aging tests can also be performed in the presence of hydrogen peroxide at elevated temperatures to predict the effect of reactive oxygen species on the electrode itself [24]. However, this effect is more relevant for metallic neural electrodes due to metal dissolution.

Therefore, in the current study, accelerated aging tests were only performed at elevated temperatures to study the degradation of HNBR. The stability of the polymer at 37 °C over 4 weeks was investigated to achieve a better understanding of its properties under physiological conditions. In addition, the HNBR-coated CNT fibers were analyzed at 60 and 70 °C for 45 days. Figure 6A–F show the morphologies of the HNBR-coated CNT fibers that were incubated at 37, 60, and 75 °C in PBS. No physical damage or delamination of HNBR was observed over 4 weeks at 37 °C in PBS (Figure 6A,B), which implies that HNBR remained unreactive during the analysis period under physiological conditions. HNBR composites have previously been reported to have excellent chemical, solvent, and aging resistance, which is attributable to the fully saturated polymer backbone structure [85].

Strong adhesion between the electrode and polymer coating material is very important for reducing water vapor accumulation between the insulating polymer and the underlying conductive components. This is a serious issue for parylene C-based systems, as parylene C shows poor adhesion with many materials and the substate must undergo surface pretreatment to increase adhesion. Notably, the excellent wetting behavior of HNBR on CNT fibers is extremely beneficial for creating seamless integration at the polymer–CNT interface. As shown in Appendix A, HNBR wets the CNT fiber surface easily, thus diminishing the risk of water vapor accumulation issues.

HNBR also remained intact without cracks, delamination, or swelling after accelerated aging at 60 and 75 °C for 45 days in PBS (Figure 6C–F). The durability of the 7 µm thick HNBR coating at these high temperatures implies that the polymer can withstand physiological conditions for much longer periods. The manufacturer (Zeon Chemicals) reports that Zetpol 2000 HNBR can withstand even higher temperatures (160 °C). For polymers such as polyimide, almost complete delamination from metal-based electrodes occurs at 87 °C over 7 days [24]. In contrast, no delamination was observed for the HNBR-coated fibers at any of the studied temperatures. However, cavity formation did occur, the degree of which was higher at 60 and 75 °C than at 37 °C. In addition, the cavities were found to form during first week with no increases observed during further incubation. Importantly, hole formation was not observed when the HNBR-coated CNT fibers were incubated at 60 and 75 °C without PBS. Therefore, this phenomenon may be due to a water-soluble impurity. An elution test was performed to analyze the possible toxic effects of water-soluble impurities, as described in Section 3.3.

EIS was performed for the HNBR-coated CNT fiber microelectrodes incubated at 37 and 60 °C to evaluate the changes in impedance and CSC over time. Generally, the formation of cracks and cavities can expose underlying conductive elements to the electrolyte solution, resulting in a reduced impedance and higher CSC_c_. The changes in impedance at 1 kHz over time for the HNBR-coated CNT fiber microelectrodes incubated at 37 and 60 °C are shown in Figure 6G,H, respectively. For each temperature condition, three electrodes are presented in Figure 6G,H. The impedance was analyzed every 7 days and the measurements were performed at room temperature. For the microelectrodes incubated at 37 °C in PBS, a gradual increase in impedance was observed over 3 weeks in the mid- to low-frequency regions (Appendix A). However, the HNBR coating remained intact, and no damage was observed. Additionally, for the microelectrodes incubated at 60 °C in PBS, an increase in impedance was observed after 2 weeks (Appendix A). However, the impedance analysis time was reduced to 2 weeks for the microelectrodes incubated at 60 °C due to the change in color (possible oxidations) that occurred at the metal/CNT fiber connections, where Ag paste was employed. Compared to electrode 1 and 2, electrode 3 showed a significant increase in the impedance at 1 kHz (Figure 6H). The SEM images of the electrodes revealed that no damage was visible at the interface of the microelectrode that was in contact with the electrolyte, but SEM images of the microelectrode surfaces revealed that debris might accumulate on the electrode surface after prolonged soaking in PBS. Therefore, the increased impedance could be due to changes in the electrode surface instead of damage to the HNBR coating. The observed CSC_c_ further confirmed the fouling of the electrode surface, in agreement with the EIS results. 

## 4. Conclusions

In this work, the CDC approach was used to coat CNT fibers with a flexible polymer, HNBR, to develop pinhole-free coatings with uniform thicknesses. Critical CDC parameters, including the HNBR concentration and withdrawal speed, were optimized to achieve uninterrupted and uniform HNBR coatings on CNT fibers. The feasibility of this technique for coating CNT fibers with diameters in the range of 25–66 µm and lengths of up to 4.8 m was successfully demonstrated. The biocompatibility of the HNBR coating was confirmed in vitro using elution tests and neuronal cultures. The microelectrodes prepared from the HNBR-coated CNT fibers showed favorable electrochemical properties for the development of neural electrodes. These electrodes achieved a specific impedance and CSC_c_ of 27.0 ± 9.4 MΩ µm^2^ and 487.6 ± 49.8 mC cm^−2^, respectively. Furthermore, aging tests revealed that the HNBR coating had better tolerance for extreme temperatures than previously reported insulating materials. These findings provide a basis for developing flexible polymer coatings on carbon-based implantable microelectrodes for long-term biomedical applications.

## Figures and Tables

**Figure 1 bioengineering-10-00647-f001:**
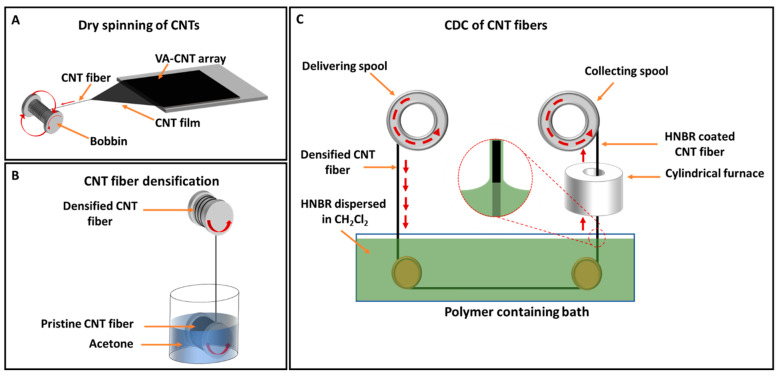
Schematic illustrations of CNT fiber assembly, densification, and polymer coating. (**A**) CNT fiber fabrication via dry-spinning of a VA-CNT array, (**B**) continuous densification of a CNT fiber using acetone, and (**C**) continuous coating of a CNT fiber with HNBR using a homemade CDC setup.

**Figure 2 bioengineering-10-00647-f002:**
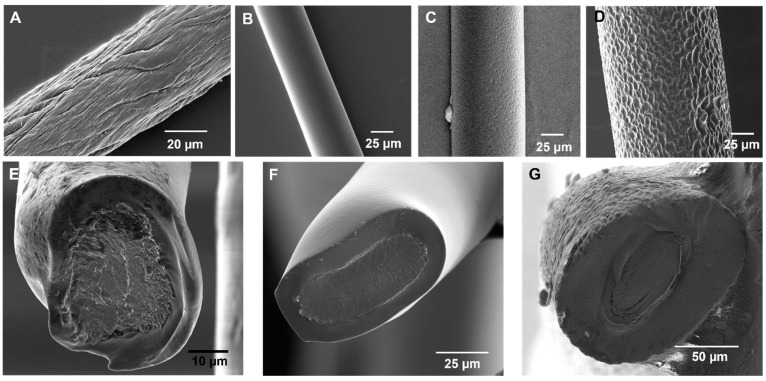
SEM characterization of uncoated and HNBR-coated CNT fibers. SEM images of (**A**) an acetone-densified CNT fiber with a diameter of 44 µm, and CNT fibers with diameters of (**B**) 24.8 ± 0.7, (**C**) 43.6 ± 1.3, and (**D**) 66.0 ± 0.3 µm after coating with HNBR using the optimal CDC parameters (0.1 g mL^−1^ HNBR dispersion in methylene chloride at a withdrawal speed of 6 mm s^−1^). Corresponding cross-sectional SEM images of the HNBR-coated CNT fibers with diameters of (**E**) 24.8 ± 0.7, (**F**) 43.6 ± 1.3, and (**G**) 66.0 ± 0.3 µm.

**Figure 3 bioengineering-10-00647-f003:**
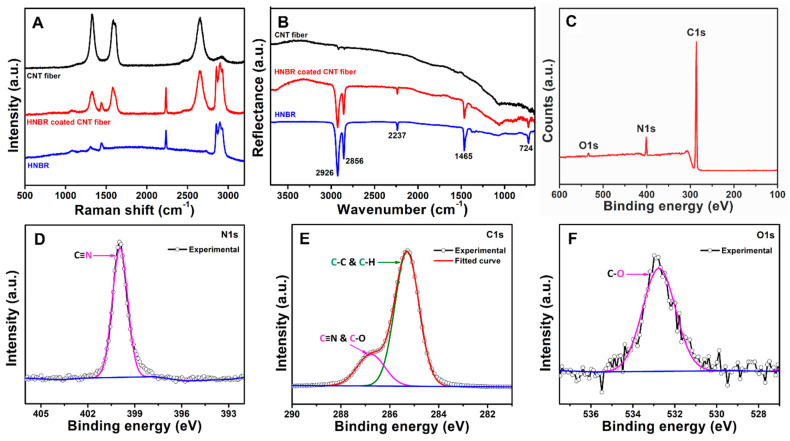
Chemical characterization of HNBR-coated CNT fibers. (**A**) Raman spectra of a pristine CNT fiber (black), an HNBR-coated CNT fiber (red), and neat Zetpol 2000 HNBR (blue); (**B**) ATR-FT-IR spectra of a pristine CNT fiber (black), an HNBR-coated CNT fiber (red), and neat Zetpol 2000 HNBR (blue); the fibers were coated using the optimal CDC parameters (0.1 g mL ^−1^ HNBR dispersion in methylene chloride at a withdrawal speed of 6 mm^−1^). (**C**) XPS survey scan and high-resolution XPS spectra at the (**D**) N 1s, (**E**) C 1s, and (**F**) O 1s core levels of an HNBR-coated glass slide; HNBR-coated glass slides were prepared as described in Section 2.5.1.

**Figure 4 bioengineering-10-00647-f004:**
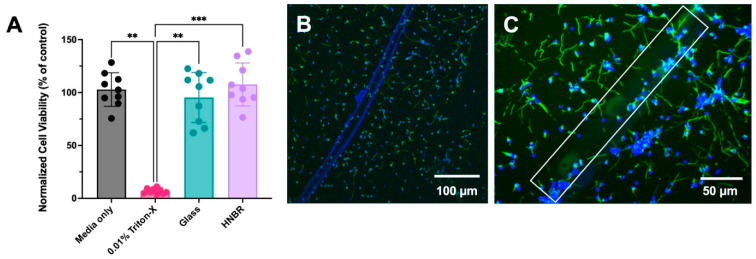
Biocompatibility of HNBR-coated samples. (**A**) XTT cell viability assay, showing no significant differences in normalized cell viability between the media obtained by soaking HNBR-coated glass coverslips for 1 day and the media-only negative control. Media with 0.01% Triton-X served as the positive control with a toxic effect on the 3T3 fibroblast cells. The data are shown as mean normalized cell viability with error bars representing the standard deviation for n = 3 cell cultures, each with 3 technical well replicates (** *p* < 0.01, *** *p* < 0.001, one-way ANOVA with Kruskal–Wallis multiple comparison test). (**B**) Image showing neuron growth on 24.8 µm HNBR-coated CNT fibers, where green and blue represent β-tubulin immunolabeling and nuclei, respectively. (**C**) Representative high-magnification image with the fiber highlighted by a white box.

**Figure 5 bioengineering-10-00647-f005:**
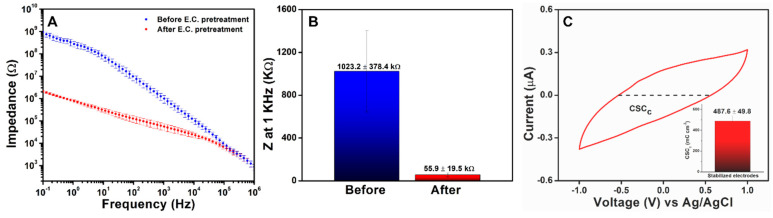
Electrochemical characterization of HNBR-coated CNT fiber microelectrodes. (**A**) Bode plot for the EIS analysis of HNBR-coated CNT fiber microelectrodes in the frequency range of 0.01–10^6^ Hz in 0.01 M PBS (pH 7.4), (**B**) impedance analysis at 1 kHz for HNBR-coated microelectrodes before and after stabilization by electrochemical pretreatment (n = 15), and (**C**) CV curve representing the CSC_c_ of an HNBR-coated CNT fiber microelectrode. CV measurements were performed in 0.01 M PBS (pH 7.4) at a scan rate of 0.1 V s^−1^. The inset show the average CSC_c_ for HNBR-coated CNT fiber microelectrodes (n = 15).

**Figure 6 bioengineering-10-00647-f006:**
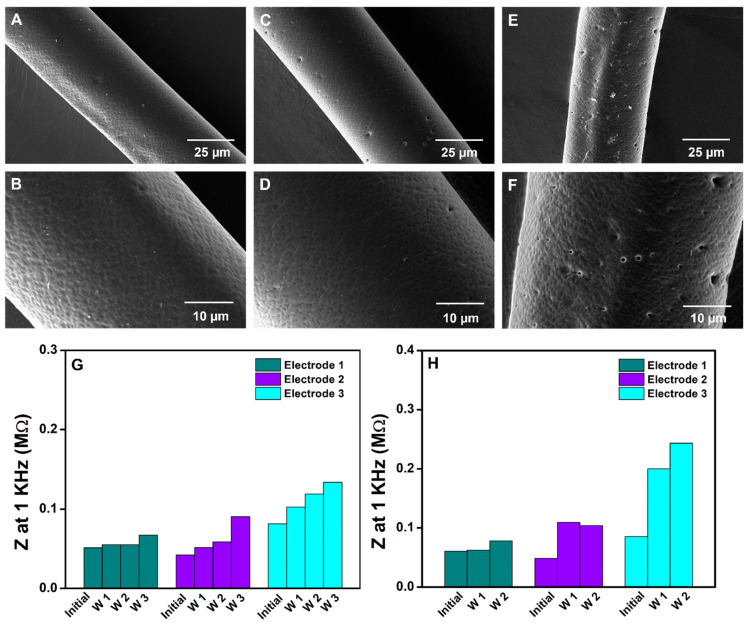
Morphological and EIS analysis of aged HNBR-coated CNT fibers. High-magnification SEM images of HNBR-coated CNT fibers after incubation (**A**,**B**) at 37 °C for 30 days in PBS, (**C**,**D**) at 60 °C for 45 days in PBS, and (**E**,**F**) at 75 °C for 45 days in PBS. EIS analysis of HNBR-coated CNT fiber microelectrodes incubated (**G**) at 37 °C for 3 weeks and (**H**) at 60 °C for 2 weeks. Weeks 1, 2, and 3 are denoted as W1, W2, and W3, respectively.

## Data Availability

Not applicable.

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
