# Peer review of "Evaluation of Polymer-Coated Carbon Nanotube Flexible Microelectrodes for Biomedical Applications"

_bioengineering, 2023, doi:10.3390/bioengineering10060647_

Round 1
Reviewer 1 Report
The submitted paper (Biocompatible ...) can be viewed from two discrete experimental research angles: 1)the materials (HNBR-coated CNTs) and 2)their biocompatibility.
The first is wonderful, but the second is, actually, unfinished, and this part should be reconsidered after major revision. The main (experimental, biocompatibility) issue(s) arrive from the a)restricted electrochemical characterization and the b)inadequate implementation of (reduced time of 10s, only) the Cu-electrodeposition. The WE (Working Electrode of the electrochemical cells) is highly anisotropic (say, structures like wood) having dimensionalities' issues similar to the scientific issues discussed in the (bibliography of the) '70s about complex mechanisms of the HOPG-intercalation compounds.
So, all about the electrochemical (and biocompatibility, Long-Term Stability, in Vitro Cytotoxicity of HNBR Coating) tests and effects are interpreted as semi-isotropic WE (and microelectrodes, HNBR-coated CNT fibers), are producing misleading 'results and discussion' in EIS (3.4.1. EIS Analysis), CV (Cyclic Voltammetry), the anisotropic Cu-electrodeposition (Figure S1), and the (anisotropic) aging (3.4.3. Aging Studies of HNBR-Coated CNT Fiber Electrodes).
Also, the (EIS) impedance Z_amplitude spectra, only, is not creditable; so, it is strongly recommended to add the Z_phase spectra (or/and the Nyquist plot) inside Figure 5(A).
Reviewer 2 Report
In this paper, the morphology, chemical nature, and insulating properties of polymer-coated carbon nanotube fibers fabricated by the continuous dip coating method were investigated. In addition, aging studies were conducted at elevated temperatures to evaluate the stability. This paper is very interesting and meaningful for long-lasting implantable neural electrodes. I recommend that this paper is published after minor revisions according to a following comments.
Comments:
1. (L295): (boiling point: -39.6 °C)---> (boiling point: 39.6 °C)
2. (Figure 6G and 6H): The authors need a description of Electrode 1-3, in particular, it should be explained that Electrode 3 has a large increase in impedance.
Round 2
Reviewer 1 Report
The manuscript of this version (v2) is better. However, it is recommended to include the word 'studies' in their title. A proposal title is: Biocompatibility studies of Polymer-Coated Carbon Nanotube Fiber Flexible
Microelectrodes.
Also, the misleading (and unsupported) value of the 'specific impedance' must be deleted, at least, from the Abstract (Line 023).
Author Response
Thanks the reviewer for this valuable suggestions. Our replies are in black to reviewer's questions in green:
it is recommended to include the word 'studies' in their title. A proposal title is: Biocompatibility studies of Polymer-Coated Carbon Nanotube Fiber Flexible
The title has been changed as suggested
Also, the misleading (and unsupported) value of the 'specific impedance' must be deleted, at least, from the Abstract (Line 023).
The specific impedance is calculated at 1.0 kHz and corresponds to measurements in Fig 5A. The details about the calculation are provided in section 3.4.1. Therefore we have clarified this information in the Abstract so we can maintain the reported impedance values